# Combined Deep Learning Techniques for Mandibular Fracture Diagnosis Assistance

**DOI:** 10.3390/life12111711

**Published:** 2022-10-26

**Authors:** Dong-Min Son, Yeong-Ah Yoon, Hyuk-Ju Kwon, Sung-Hak Lee

**Affiliations:** 1School of Electronic and Electrical Engineering, Kyungpook National University, 80 Daehakro, Buk-gu, Daegu 41566, Korea; 2School of Dentistry, Kyungpook National University, 2177 Dalgubeol-daero, Jung-gu, Daegu 41940, Korea

**Keywords:** mandibular fracture, panoramic radiography, image processing, object detection, deep learning, YOLO, Mask R-CNN, U-Net

## Abstract

Mandibular fractures are the most common fractures in dentistry. Since diagnosing a mandibular fracture is difficult when only panoramic radiographic images are used, most doctors use cone beam computed tomography (CBCT) to identify the patient’s fracture location. In this study, considering the diagnosis of mandibular fractures using the combined deep learning technique, YOLO and U-Net were used as auxiliary diagnostic methods to detect the location of mandibular fractures based on panoramic images without CBCT. In a previous study, mandibular fracture diagnosis was performed using YOLO learning; in the detection performance result of the YOLOv4-based mandibular fracture diagnosis module, the precision score was approximately 97%, indicating that there was almost no misdiagnosis. In particular, fractures in the symphysis, body, angle, and ramus tend to be distributed in the middle of the mandible. Owing to the irregular fracture types and overlapping location information, the recall score was approximately 79%, which increased the detection of undiagnosed fractures. In many cases, fractures that are clearly visible to the human eye cannot be grasped. To overcome these shortcomings, the number of undiagnosed fractures can be reduced using a combination of the U-Net and YOLOv4 learning modules. U-Net is advantageous for the segmentation of fractures spread over a wide area because it performs semantic segmentation. Consequently, the undiagnosed case in the middle of the mandible, where YOLO was weak, was somewhat supplemented by the U-Net module. The precision score of the combined module was 95%, similar to that of the previous method, and the recall score improved to 87%, as the number of undiagnosed cases was reduced. Through this study, the performance of a deep learning method that can be used for the diagnosis of the mandibular bone has been improved, and it is anticipated that as an auxiliary diagnostic inspection device, it will assist dentists in making diagnoses.

## 1. Introduction

Mandibular fractures are a type of fracture treated in dentistry that are commonly caused by accidents [1]. Patient opinion, panoramic radiographic images, and CBCT (cone-beam computed tomography) images are used for the diagnosis of fractures. Panoramic radiography and CBCT are mainly used to confirm the exact size and location of the fracture during the diagnosis of the patient’s painful area [2]. Fracture locations that are difficult to identify in a panoramic radiographic image are identified through a CBCT image. Deep learning methods using CT are also being studied and it can be confirmed that they have high accuracy [3]. However, because CBCT takes a long time to perform and is more expensive than panoramic radiographs [4,5], this study introduces image deep learning technology to help determine the location of fractures from panoramic radiographic images. This helps the doctor to make a visual diagnosis and can assist in the identification while conserving time and cost. On panoramic radiographic images, the fracture site can be divided into six parts based on anatomical standards [6]. The fracture site is divided into the symphysis, body, angle, ramus, condyle, and coronoid, the most common of which are the symphysis, body, and angle (Figure 1) [1].

The shape of fractures in the panoramic radiographic image generally shows an oblique fracture with a gap and a shear fracture with a step in red boxes one and two (Figure 2). However, in the condyle region, the fracture shape differs from the general fracture shape, and the bones in the condyle region seem to overlap (dislocation) in red boxes 3 and 4 [7]. Owing to the similarity between the phenomenon that looks like a fracture due to the difference in shading on the panoramic radiographic image and the shape of a specific fracture, such as a fracture in the condyle region, the expert needs to make a careful judgment. Therefore, deep learning methods can help reduce physician errors and support medical diagnoses.

Nishiyama et al. proposed a method for diagnosing mandibular condyle fractures that detected only fractures in the condyle region using a deep learning technique with panoramic radiographic images collected from two hospitals [8]. The deep learning network used in this study is AlexNet, CNN-based DIGITS Library v 5.0. Five convolutional layers and three fully connected layers were used, based on the Caffe framework provided by the DIGITS library. Different modules were presented for a total of three data sets: A, B, and A + B hospital datasets; among them, the score of the module based on the A + B hospital dataset showed the best result of approximately 85% based on the sensitivity (precision score). However, this module has the disadvantage of being able to diagnose only condyle fractures among mandibular fractures and has a high rate of false diagnoses for fractures.

Son et al. presented a method for diagnosing mandibular fractures through data augmentation using the LAT (luminance adaptation transform) image processing method and YOLOv4 (LAT YOLOv4) [9]. The fracture area was divided into six parts according to anatomical standards, and learning was performed. YOLO is a one-stage object detection technique with an object detection learning frame that simultaneously performs classification to identify and localize an object for determining its location [10,11]. YOLO detects mandibular fractures in a box shape and has a good learning ability for location information. Panoramic radiographic images of mandibular fractures have different resolutions of patient images for learning and diagnosis; however, the side area is compressed in the left and right directions compared to the center of the panoramic radiographic image. Therefore, the non-detection rate of fractures in the central region is increased in the YOLO learning results.

Mask R-CNN is a two-stage object detection technique with an object detection learning frame structure in which region proposals and classifications are sequentially performed [12]. The two-stage object detection learning technique generally is more accurate but slower than the one-stage detection. The mask R-CNN model requires a total of three steps: region proposal, classification, and bounding box regression. Moreover, the learning pipeline structure is complicated owing to separate and individual learning. The structure of Mask R-CNN additionally includes a branch predicting a unit segmentation mask for the region of interest (RoI) of faster R-CNN. The inference time occurs because learning proceeds sequentially. As a result of learning, Mask R-CNN tends to judge gaps between teeth as fractures, and overall, its ability to detect mandibular fractures is inferior to that of the LAT YOLOv4 technique.

U-Net is a biomedical segmentation model [13] that is widely used in the medical field. Unlike YOLO and Mask R-CNN, U-Net is a technique used for image segmentation rather than object detection. U-Net can prevent the loss of location information to a certain extent by adding a long skip connection to the layer in the early and latter parts [14]. To detect mandibular fractures, U-Net learns by labeling fractures. However, it has a disadvantage as it cannot adequately detect fractures in the dislocated fracture area of the condyle, which is the side area. Despite this, because U-Net detects mandibular fractures in the middle part, AACZXait can compensate for the undetected area of the LAT YOLOv4 to some extent.

In this study, to improve the classification of fracture types, which is a problem in the existing mandibular fracture detection using YOLOv4, segmentation of the fracture line was performed based on U-Net. The learning area was divided into lines according to the fracture shape. YOLO detects fractures in a box shape; however, U-Net has the advantage of segmenting fractures more accurately because it performs line semantic segmentation, among fractures in the condyle region, there are generally many dislocated fractures, and segmentation for learning is difficult. If the fracture is segmented and trained forcibly in consideration of the dislocation, adverse effects may occur in the learning result. Therefore, U-Net was applied only to fracture segmentation learning for the rest of the mandible fracture, except for the side region (condyle and coronoid process). As the YOLO module cannot be detected, it is possible to detect the fractured part of the undiagnosed area to some extent through U-Net segmentation. In the experimental results, the method in which the two learning modules were merged showed complementary improved performance by applying YOLO to areas that U-Net could not detect well. Recall and F1 scores increased compared to models using only the existing YOLO technique, enabling better fracture detection. Fractures in the rest of the region, except for the side region, can be segmented well to some extent.

## 2. Materials and Methods

### 2.1. Data Segmentation

YOLO, Mask-R-CNN, and U-Net use different data-segmentation methods when constructing the training data. First, the mandibular fracture diagnosis module using Son et al.’s LAT YOLOv4 has a good diagnostic ability for fracture positions; however, it has a high undetected rate because it does not accurately distinguish fracture shapes and expresses fracture areas in the form of boxes. This module shows the diagnostic performance of precision of 97%, recall of 79%, and F1 score of 88%. In particular, fractures in the symphysis, body, angle, and ramus tend to be distributed in the middle of the mandible, and undiagnosed detection errors increase owing to irregular fracture shapes and overlapping location information. The advantage of location information is weakened; there is the possibility of a case where a fracture that is visible to the naked eye cannot be detected (Figure 3b). Therefore, additional applications such as Mask R-CNN, which is another object detection deep learning, and U-Net, which is used for medical image region segmentation, are needed to improve detection capabilities, especially recall scores representing undiagnosed performance in existing studies.

Unlike YOLO learning data consisting only of box coordinates and class information, Mask R-CNN requires the designation of a polygon-type learning region in the fracture region for the construction of a mandibular fracture learning dataset. However, in the case of shear fractures without gaps or dislocation fractures occurring in the condyle region, it was difficult to express the training data region using Mask R-CNN. Because these specific fracture regions cannot be drawn in closed polygon form, fracture regions are represented and trained in a box or simple form, such as YOLO learning data. In this case, for comparison with the results of the LAT YOLOv4 module [9], training was performed with the same class structure on the same training data, and the detection results in the same test data were 57% precision, 70% recall, and 63% F1 scores. The fracture shape of the condyle region is not properly expressed, which leads to an error in judgment owing to the fracture shape, and the error detection between teeth is also recognized as a fracture in the panoramic image, resulting in a lower precision score (Figure 3c).

For U-Net training, a training dataset labeled according to the shape of the fracture is constructed. This makes it easy to generate U-Net training datasets of fracture sites with complex spacing. However, shear fractures or displaced fractures are difficult to be labeled. Therefore, in the case of shear fractures, the parts which are visible even with solid and crack lines in the shear fractures were labeled as much as possible. But the displaced fractures were not labeled because it adversely affected the fracture detection accuracy during U-Net training. Additionally, when segmentation and labeling are performed differently from complex fracture shapes, it adversely affects the training process and reduces mandibular fracture detection performance. Figure 3d confirms that U-Net shows good segmentation performance for general fracture types but U-Net shows poor segmentation performance for dislocated fractures in the condyle region because U-Net was not trained in condyle region. Therefore, U-Net is advantageous for learning the rest of the fracture areas except for some of the severe shear and displaced fractures.

### 2.2. U-Net Architecture

U-Net is a deep learning network model for image segmentation, which is mainly used in the biomedical field and is widely applied in various fields such as the vision of autonomous vehicles (road detection, pedestrian detection). This convolutional neural network is expressed as a U-Net because it is composed of convolutional layers and max pooling in a U-shape (Figure 4). U-Net can be divided into two major paths: contracting and expanding. The contracting path extracts meaningful information from a large input image. The expanding path performs up-sampling by combining (skip-connection) contextual information extracted from the contracting path and the location information of the pixels existing in each layer. U-Net is an encoder-decoder-based model such as an autoencoder [15]. In general, in the encoding step, the dimension is reduced as the number of channels increases to effectively express the characteristics of the input image. In the decoder step, the encoded information undergoes dimensional reduction, and the detailed location information of the object in the image is lost. Therefore, in the expanding path stage, U-Net has a network structure to maintain high spatial frequency information by adding a skip connection and attempts to solve the shortcomings of the encoder-decoder. The standard U-Net uses the structure of a 3 × 3 convolutional layer, batch normalization, and active function ReLU (rectified linear unit). Various activation functions, such as ReLU, Leaky ReLU, and ELU (exponential linear unit), have been recently provided, and different performance results appear depending on the learning characteristics. This study used ELU instead of ReLU to reduce the false detection rate for the line detection of mandibular fractures. Unlike ReLU, ELU uses an exponential function when the input is negative; therefore, for an input smaller than 0, the differential value does not become zero, and the dying ReLU problem does not occur [16].

The performance scores for the three different activation functions are compared in Table 1. In the experiment, a training module was created using the same 360 fracture panoramic images as the training data, and the results were compared with the test data of 60 fractional panoramic images. ELU can reduce the misdetection rate (precision) compared to other activation functions; however, it tends to increase the ‘undetection’ rate (recall). However, considering the average score (F1 score), the ELU activation function yielded the best score. Therefore, in this study, the U-Net network activation function s trained using ELU instead of ReLU.

### 2.3. Proposed Deep Learning Module

#### 2.3.1. Training Data Augmentation

Data augmentation significantly improved the accuracy of the module during training. In the case of the LAT YOLOv4 module, the detection performance was improved by increasing the number of training data images using LAT and gamma correction, which are image pre-processing and data augmentation provided by default. LAT processes images in two ways: single-stage luminance adaptation transform (SLAT) and multiple-stage luminance adaptation transform (MLAT) [17]. As shown in Figure 5, SLAT strongly enhances the local details of the panoramic image; therefore, the image becomes brighter overall. Compared with SLAT, MLAT improves the visual performance of an image through tone compression of the entire image rather than local details. The LAT-processed panoramic image emphasizes the mandibular fracture better than the conventional image.

Additionally, the YOLO-based module classified mandibular fractures into six classes (symphysis, body, angle, ramus, condyle, and coronoid process) according to anatomical location and conducted the training. Table 2 shows the comparison scores when using the standard YOLOv4 module and the module with LAT processing added. Because LAT image data augmentation can improve both misdetection and ‘undetection,’ LAT image pre-processing was used for learning mandibular fracture detection.

The LAT YOLOv4 module has a good ability to diagnose the location of fractures; however, the fracture detection performance is poor in the symphysis, body, angle, and ramus parts of the mandibular fracture because the resolution of each panoramic image is different; a patient’s movement during radiography may produce blurred areas in the image and large step defects appear in the inferior border of the mandible. Furthermore, some proximal areas and the column of the ramus may overlap [18]. Therefore, the YOLO learning module cannot detect even a part of the target area in which it would be well-positioned to diagnose a fracture. To supplement the middle region of the mandibular fracture, which the LAT YOLOv4 module cannot diagnose well, U-Net’s biomedical segmentation characteristic was used to perform line segmentation, which is close to the fracture shape on the middle fracture part. Additionally, flipping and random cropping data augmentation were used to increase the learning performance of the U-Net during training. Table 3 shows the results of adding random flip and random crop augmentation to U-Net. It is concluded that the overfitting problem has been significantly improved to some extent through random crop augmentation. The random crop performs data augmentation by cropping approximately 80% of the input network image (512 × 512) based on the random coordinates at the top left.

#### 2.3.2. Training and Combined Detection Process

In this study, we propose a combined module that detects fractures in the entire mandible by detecting the symphysis, body, angle, and ramus using the U-Net module and condyle and coronoid regions using the LAT YOLOv4 module. The U-Net module is advantageous in fracture line detection in the symphysis, body, angle, and ramus regions. The LAT YOLOv4 module has excellent detection performance in the condyle and coronoid processes. This module consists of two parts for detecting mandibular fractures: fracture and tooth-line region learnings.

Figure 6 shows that the first learning process is to segment the mandibular fracture line by designating 360 mandibular fracture panoramic images and 360 fracture line segmented labeling images as learning datasets. In the test process in Figure 6, the mandibular fracture U-Net module uses panoramic images from a set of 60 test data to create a mandibular fracture label result image.

The second U-Net learning method is the tooth line area learning method. The training data were formed by pairing 60 training data panorama images and 60 tooth line area label images, and learning was performed using U-Net (Figure 7). Initially, to learn tooth line area detection, tooth line area detection learning was performed with 30 training datasets. Consequently, there was a hole in the tooth line area or the tooth line area was not properly marked. To correct these shortcomings, 30 images of data were added to the training data to increase the accuracy of tooth line area detection, and 60 pieces of data were used to conduct tooth line area detection learning. As shown in Figure 8, it can be observed that the performance of the module trained with 60 images of data is better than the tooth line detection result with 30 images of data. In Figure 9, the result of the tooth line area label obtained by the tooth line U-Net module is filled with a hole in the tooth line area through dilation to generate the final tooth line label image. If the fracture line of U-NET existed in the tooth line area, the resulting image of the tooth line area module was inverted and used as a weight map to erase the false detection area of the fracture line in the tooth line area. The reason for learning the tooth line area is that when the fracture label exists in the tooth line area in the first U-Net module, the error rate increases; therefore, the tooth line area module is trained to use it as a weight map of the fracture label.

Figure 10 illustrates the overall process of the proposed mandibular fracture detection module. First, the tooth line area was detected through the tooth line area U-Net module, and the tooth line area label was inverted to change the tooth line area part to a black label. Second, to extract the fracture line, a fracture line label image was generated through the module trained only with the fracture line label. To reduce the erroneous detection of the fracture line segmentation of U-Net, the inverse tooth, and fracture line label image were multiplied to eliminate the misdetection fracture line existing in the tooth line region. By using the tooth line area, the fracture line label from which the misdetected fracture line has been removed is combined with the original panoramic image to obtain the final result image of the U-Net fracture module. Subsequently, the final mandibular fracture diagnosis result image was created by combining the box-shaped fracture detection area detected by the YOLOv4 LAT module and the results of the U-Net fracture result module.

## 3. Results

### 3.1. Deep Learning Networks and Dataset

Deep learning methods were implemented on a PC with an Intel i7-9700K processor, 32 GB RAM, and an NVIDIA TITAN RTX. The deep learning networks consisted of the Windows version of YOLOv4 and U-NET using PyTorch. The training dataset consisted of 360 panoramic radiographs of mandibular fractures, and the test dataset consisted of 60 panoramic radiographs of mandibular fractures. The resolution of the radiographs ranged from 2288×1244 to 2972×1536 pixels. All datasets were approved by the Institutional Review Board (IRB) of Kyungpook National University Dental Hospital.

Among the 420 mandibular fracture panoramic radiographs, 360 training data images are selected, which are good to determine fractures, and other images are used as test images. Training data and test data are completely separated. Mandibular fractures are classified into six anatomical structure types. Symphysis, body, angle, and ramus are expressed as middle fractures in this study, and condyle and coronoid processes are expressed as side fractures. Generally, when the fracture distributions of the training data and the test data are divided into middle and side fractures, the distribution is generally similar as shown in Table 4 and Table 5.

Also, in Dongas, P. et al. [1], it is divided into 7 regions as symphysis, body, angle, ramus, subcondyle, condyle, and coronoid. In that case, middle fractures (symphysis, body, angle, ramus, subcondyle) account for 90.4% and side fractures (condyle, coronoid) account for 9.6%. In our study, ramus and subcondyle are collectively referred to as ramus and classified into 6 regions. Also, the middle fractures of 91.666% and the side fractures of 8.334% for the training data, the middle fractures of 84.536% and side fractures of 15.084% for the test data are nearly consistent with the referred case. Therefore, it can be seen from the fracture distribution of training and test data that the deep learning method of this study is close from general fracture distributions.

### 3.2. Evaluation Metric

The evaluation metric is an important value that can be used to evaluate the performance of deep-learning modules. In this study, performance was compared using three evaluation indicators: precision, recall, and F1 scores.
(1)$$Precision=\frac{\mathrm{Fracture}\;\mathrm{detection}\;\left(\mathrm{True}\;\mathrm{Positive}\right)}{\mathrm{Fracture}\;\mathrm{detection}\;\left(\mathrm{True}\;\mathrm{Positive}\right)+\mathrm{Fracture}\;\mathrm{Misdetection}\;\left(\mathrm{False}\;\mathrm{Positive}\right)},$$
(2)$$Recall=\frac{\mathrm{Fracture}\;\mathrm{detection}\;\left(\mathrm{True}\;\mathrm{Positive}\right)}{\mathrm{Fracture}\;\mathrm{detection}\;\left(\mathrm{True}\;\mathrm{Positive}\right)+\mathrm{Fracture}\;\mathrm{Undetection}\;\left(\mathrm{False}\;\mathrm{Negative}\right)},$$
(3)$$F1\;score=\frac{2\times \mathrm{Recall}\;\times \mathrm{Precision}}{\mathrm{Recall}+\mathrm{Precsion}},$$

The precision score is related to the misdetection of the fracture diagnosis, as shown in Equation (1); if the number of misdetections in the fracture diagnosis is small, the precision score increases. The recall score is related to the ‘undetection’ of the fracture diagnosis, as shown in Equation (2), and if there is less detection, the recall score increases. Therefore, it is not possible to claim that the performance of either module is better if the precision or recall scores are higher; fortunately, though, it is possible to compare the precision and recall scores through the F1 score to determine whether the performance is improved. The F1 score is the harmonic average of precision and recall, and the performance evaluation metric of the module can be determined using the F1 score. There is no true negative in this evaluation metric because there is no normal patient image in the test panorama image first, and it is not correct to use the true negative metric as it does not know where the fracture occurs.

### 3.3. Comparison of Deep Learning Modules

We compared the fracture detection capabilities of Mask R-CNN, YOLOv4, U-Net, LAT YOLOv4, and U-Net with LAT YOLOv4 using the same 360 training datasets and 60 test sets and showed that the proposed U-Net with LAT YOLOv4 module has improved mandibular fracture detection capabilities over other modules. First, the Mask R-CNN module was classified into six classes (symphysis, body, angle, ramus, condyle, and coronoid) on an anatomical basis and the fracture region was marked in the form of a polygon. Condyle fractures have many dislocated fractures; therefore, the fracture cannot be marked in the form of polygons and was marked as box-shaped. The learning was performed only with Mask R-CNN basic augmentation, without any other special data augmentation. The training data of the YOLOv4 module was also classified into six classes, such as the case of Mask R-CNN, and learned using YOLOv4 basic augmentation. The U-Net module did not have class classification and labeled mandibular fractures to train 360 panoramic images of the same training dataset. The LAT YOLOv4 module is a combination of MLAT and SLAT modules, and the YOLOv4 module was trained by processing MLAT, SLAT, and Gamma correction images on training data images, and each of the MLAT and SLAT modules was trained with 1080 panoramic images of training data. The U-Net and LAT YOLOv4 modules are a combination of U-Net and LAT YOLOv4 modules, and U-Net modifies the existing activation function to ELU. Because of the dislocated fracture of the condyle, only the remaining fracture areas, except for the condyle and coronoid, that is, the side fracture area was intensively trained. The training data used in this module also used the same 360 panoramic images of training data as the previous learning module. The final proposed mandibular fracture detection module is a combination of the LAT YOLOv4 and U-Net modules.

In Table 6, the parameters used for training are indicated, and in Figure 11, Figure 12 and Figure 13, the results of the doctor’s diagnosis, Mask R-CNN, YOLOv4, U-Net, LAT YOLOv4, and U-Net withYOLOv4 are compared. Figure 11 illustrates fractures in the angle and condylar regions, and in Mask R-CNN, the symphysis is misdiagnosed as a fracture. In Figure 12, the Mask R-CNN misdiagnoses the symphysis as a fracture, the result of the fact that the YOLOv4 module has better performance detecting angle fractures rather than does the LAT YOLOv4, as shown in Figure 12c,e. In short, the LAT-processed image does not always have the advantage of better revealing fractures compared with the normal panoramic radiograph.

For the case of Figure 12 images, it is possible to check the amount of local contrast improvement from the line profile information. In order to compare the changes in pixel brightness near angle fracture in the normal, SLAT and MLAT panoramic radiographs, the result of line profiles are shown in Figure 14 and Table 7. When comparing the maximum pixel brightness, minimum pixel brightness, average pixel brightness, and standard deviation in the line profile of the straight arrow near the angle fracture site, the normal panoramic radiograph has the largest standard deviation value of 5.6. Due to the characteristics of LAT processing, a dark area increases contrast, but as it becomes a bright area, contrast is maintained or slightly lowered.

However, except for some bright radiograph images, the effect of LAT processing is evident in most dark radiograph images. As shown in Figure 15, most of the LAT-processed radiographs have a high standard deviation value compared to the normal radiographs. In Table 7, the LAT-processed panoramic radiographs shows the higher deviation values of 14.6 and 15.2, which are reasonably more than 9.3 of the normal panoramic radiograph.

In Figure 13, the ramus is misdiagnosed as a fracture. Therefore, Mask R-CNN had the lowest precision score because the misdiagnosis rate was higher than that of the other modules (Figure 16). The YOLO module has a low misdiagnosis rate, while the ‘undiagnosis’ rate is high; therefore, the F1 score is low owing to the ’undiagnosis’ rate. As shown in Figure 11, Figure 12 and Figure 13, YOLOv4 and LAT YOLOv4 have strong advantages over location information; therefore, they tend to detect well in the condyle region, that is, the side fracture, while they tend to detect poorly in the symphysis, body, and angle regions, where location information is ambiguous. Unlike Mask R-CNN and YOLO, U-Net is an image segmentation deep learning network, not an object detection deep learning network and labels mandibular fractures. U-Net marks fractures as lines on the label during training; however, it is difficult to label dislocated fractures, such as condyle fractures (Figure 11). Therefore, in the U-Net module, the side fracture was not diagnosed or misdiagnosed, and the precision-recall score was lower than that of the YOLO modules. It was judged that if the two deep learning networks are used together, the shortcomings of YOLO and U-Net complement each other and help improve mandibular fracture performance. In the proposed module, duplicate boxes that occurred in LAT YOLOv4 were removed before merging with the U-Net. In the proposed U-Net with LAT YOLOv4, the precision score was reduced; however, many ‘undiagnoses’ were eliminated; therefore, the recall score was increased, and it can be observed that the overall F1 score improved the performance by more than 90%.

## 4. Discussion

Panoramic images of mandibular fractures have different resolutions and patient positions when taking pictures; therefore, there is a limit to diagnosing mandibular fractures using panoramic images. This leads doctors to diagnose mandibular fractures using CT to accurately determine the fracture location. This study primarily aims to diagnose mandibular fractures using only panoramic images to save money and time and to help doctors diagnose mandibular fractures as an auxiliary device.

Deep learning networks such as Mask R-CNN, YOLOv4, and U-Net have been used to detect mandibular fractures using only panoramic images. The advantages of the three networks can be identified experimentally through panoramic mandibular fracture images. First, Mask R-CNN specifically marks the fracture area; however, it also marks fractures in areas that are not fractured, such as dark shaded areas in the panoramic image or gaps between teeth; therefore, the false diagnosis rate is higher than that in other deep learning networks. In YOLO, the location information for the six classes divided into anatomical structures is helpful to detect mandibular fractures. However, there are some undiagnosed fracture areas, other than the fractures of the condyle and coronoid process with clear characteristics and location. Unlike the above two deep learning networks, U-Net performs training by labeling fracture and panoramic images. When creating training fracture labeling data, the dislocated fracture area of the condyle fracture is difficult to label. Unlike YOLO, the side fracture area is weak, causing an increase in the misdiagnosis rate when learning by labeling a dislocated fracture. In Figure 17, the mandibular fracture detection results images of LAT YOLOv4 and U-Net with LAT YOLOv4 are compared. The panoramic images in the first and second rows demonstrate the advantages of YOLO and U-Net. While YOLO detects condyle dislocation fractures well, it does not detect angle fractures; however, U-Net detects angle fractures well. The panoramic images in the third and fourth rows show that the undiagnosed area is reduced by using U-Net for the angle area that the LAT YOLOv4 module does not diagnose. Because U-Net performs semantic segmentation, it is advantageous for the segmentation of fractures spread over a wide area. Consequently, the undiagnosed case in the middle of the mandible, where LAT YOLOv4 is weak, is somewhat supplemented by the U-Net module.

Also, LAT processed images need to be selectively used according to the characteristics of fracture during training. A general improvement can be expected in a dark X-ray image, but there is a possibility that the LAT-processed image may not be helpful for learning in the case of an overall bright image or a fracture of a bright region. When augmenting training images, selective application of LAT images is required after profiling the fracture region.

Finally, using the test data, the estimation time for the mandibular fracture is compared with the LAT YOLOv4 module and the U-NET with LAT YOLOv4 module in Table 8. Compared with the LAT YOLOv4 model, the proposed model takes about 4 times longer. The proposed model takes about half-second for one radiograph for fracture detection. Considering the general time-to-diagnosis, this is an acceptable time delay. Therefore, it might be helpful for dentists and radiologists to detect the fractures.

## 5. Conclusions

A deep learning network was used to diagnose mandibular fractures with panoramic images, and a different object detection called Mask R-CNN was used to improve mandibular fracture detection performance in the previous study instead of LAT YOLOv4; however, the high misdiagnosis rate in Mask R-CNN did not benefit the previous study. U-Net, an image segmentation deep learning model was used. When U-Net was used alone, there was a weakness in the segmentation of dislocated fractures in the condyle region; therefore, the performance was lower than that of the LAT YOLOv4 module. Among the characteristics of panoramic images, one disadvantage is that the resolution of the image varies, and side regions are compressed in the left and right directions, compared with the center of the panoramic image. However, because the location information overlaps with the irregular fracture shape, the advantage of fracture detection using YOLOv4 location information tends to decrease. U-Net is advantageous for the segmentation of widely spread fracture areas, and consequently, U-Net is used together to supplement the undiagnosed cases of mandibular fracture, an area where LAT YOLOv4 is weak. This scheme improved the mandible fracture diagnosis rate by more than 90%, based on the F1 score. Also the proposed method diagnoses mandibular fractures in panoramic images through deep learning, for 0.522 s per one panoramic image, reducing the cost and time, and this might suggest that it can be used as an auxiliary device for doctors. By applying the deep learning technique in this study, the goal was to enable deep learning to diagnose a complete mandibular fracture by itself.

## Figures and Tables

**Figure 1 life-12-01711-f001:**
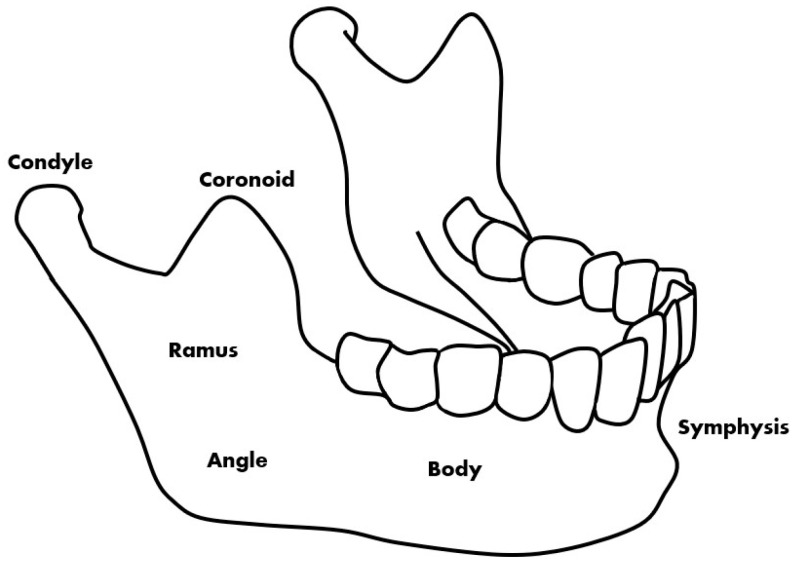
The types of mandibular fractures by region.

**Figure 2 life-12-01711-f002:**
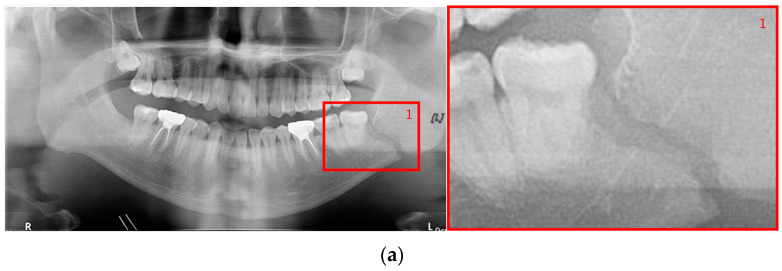
The shapes of mandibular fractures: (**a**) oblique fracture on an angle area; (**b**) shear fracture and severe shear fracture on a symphysis area; and (**c**) displaced fracture on a condyle area.

**Figure 3 life-12-01711-f003:**
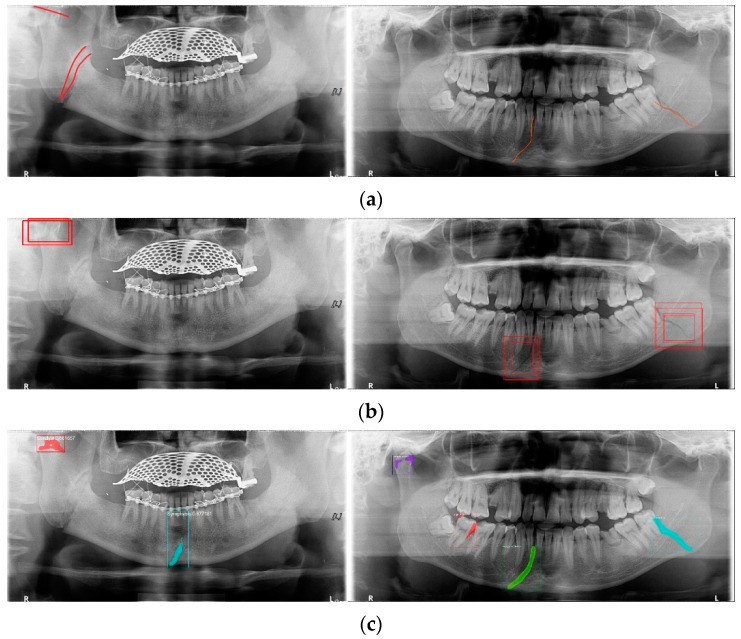
The result of mandibular fracture detection: (**a**) diagnosis by oral and maxillofacial radiologist, (**b**) by LAT YOLOv4 module (with red boxes), (**c**) by Mask R-CNN (with colored polygons and boxes), and (**d**) by U-Net.

**Figure 4 life-12-01711-f004:**
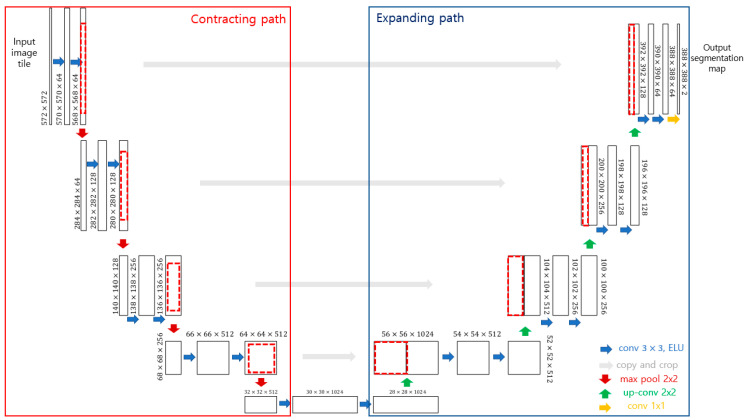
The U-Net architecture [13].

**Figure 5 life-12-01711-f005:**
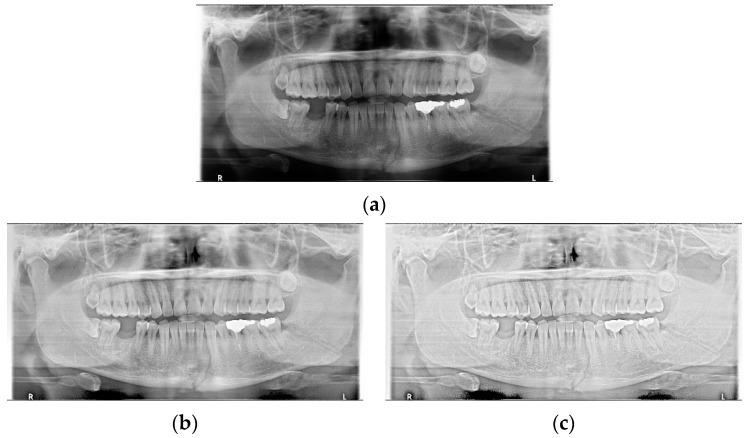
The panoramic radiographs: (**a**) normal panoramic radiograph, (**b**) multiple-stage luminance adaptation transform (MLAT) panoramic radiograph, and (**c**) single-stage luminance adaptation transform (SLAT) panoramic radiograph.

**Figure 6 life-12-01711-f006:**
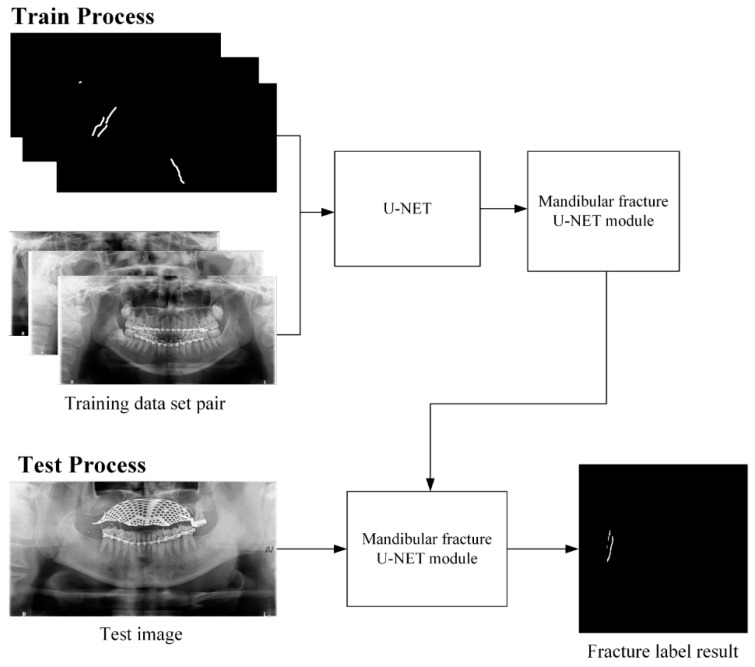
U-Net training and testing process for detecting mandibular fracture line.

**Figure 7 life-12-01711-f007:**
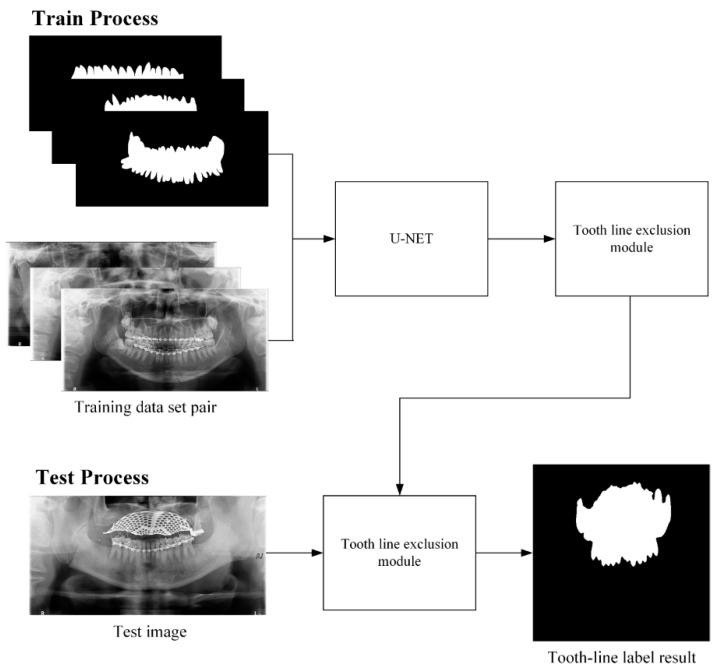
U-Net training and testing process for generating tooth line area.

**Figure 8 life-12-01711-f008:**
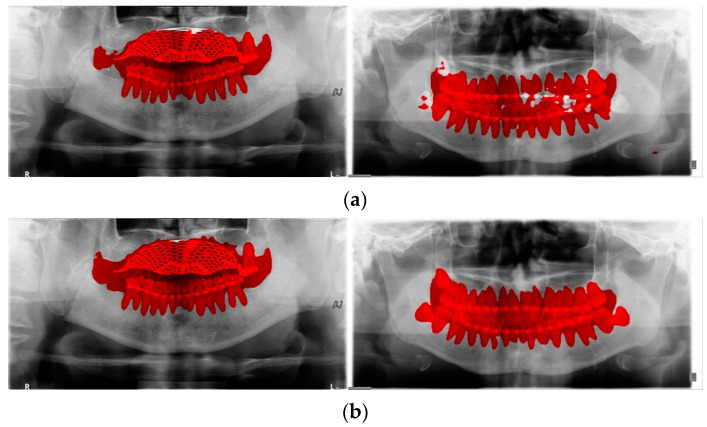
Tooth line area labeling by U-Net: (**a**) results using 30 and (**b**) 60 panoramic radiographs.

**Figure 9 life-12-01711-f009:**
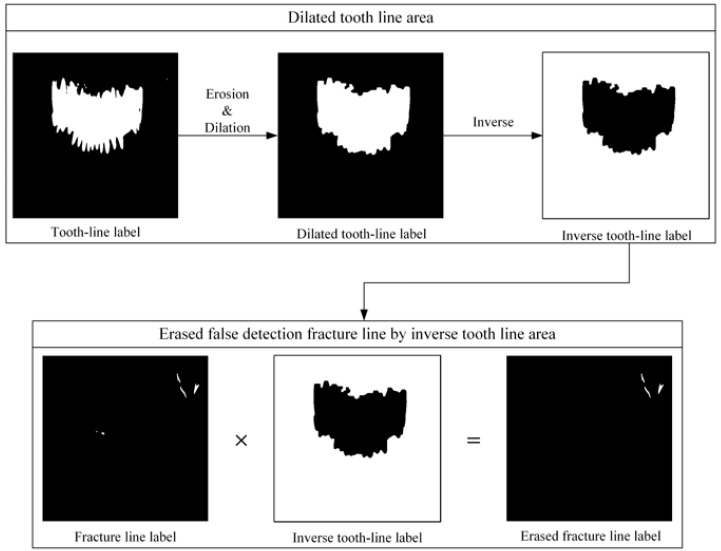
Removal of false-detection fracture line using an inverse tooth-line label.

**Figure 10 life-12-01711-f010:**
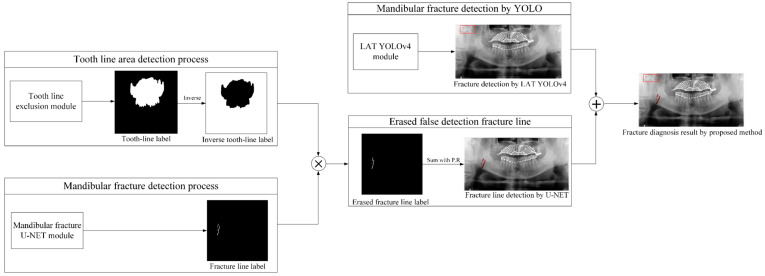
The proposed block diagram of the combined U-Net withYOLOv4 mandibular detection module (red box detected by YOLO, red line detected by U-Net).

**Figure 11 life-12-01711-f011:**
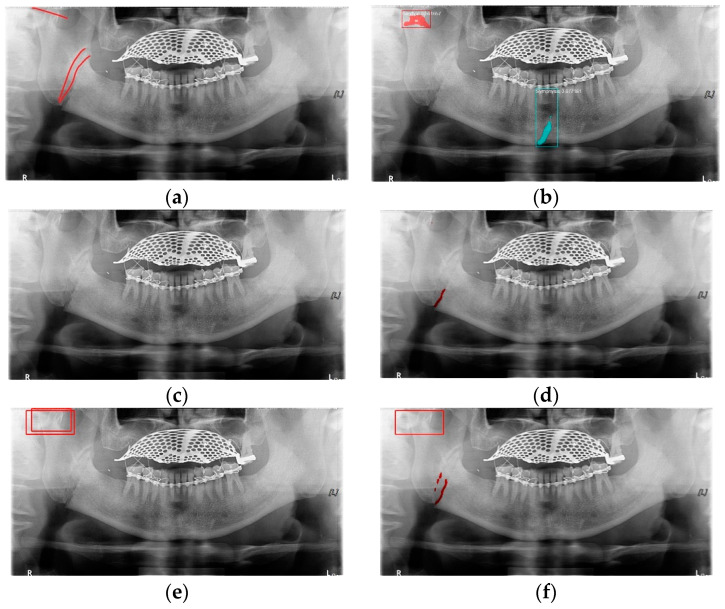
The comparison of mandibular fracture detection: (**a**) oral and maxillofacial radiologist, (**b**) Mask R-CNN (with colored polygons and boxes), (**c**) YOLOv4, (**d**) U-Net (with a red line), (**e**) LAT YOLOv4 (with red boxes), and (**f**) U-Net with LAT YOLOv4 (with a red box and lines).

**Figure 12 life-12-01711-f012:**
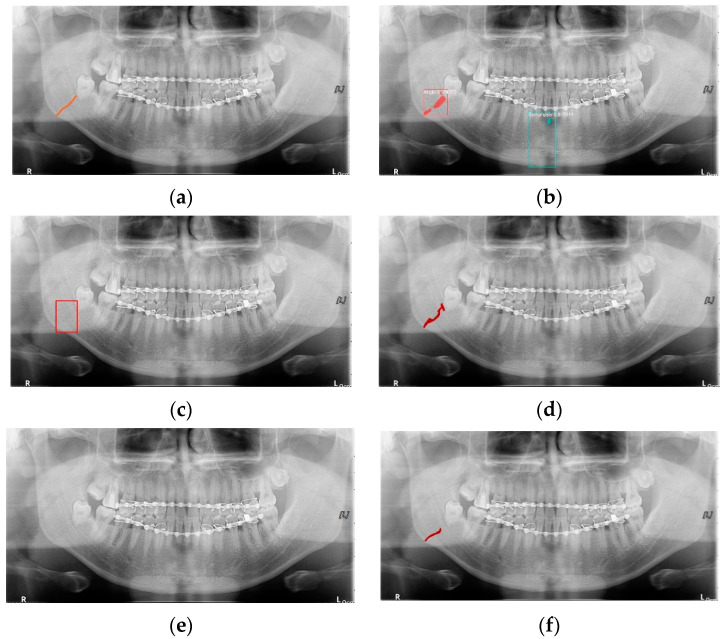
The comparison of mandibular fracture detection: (**a**) oral and maxillofacial radiologist, (**b**) Mask R-CNN (with colored polygons and boxes), (**c**) YOLOv4 (with a red box), (**d**) U-Net (with a red line), (**e**) LAT YOLOv4, and (**f**) U-Net with LAT YOLOv4 (with a red line).

**Figure 13 life-12-01711-f013:**
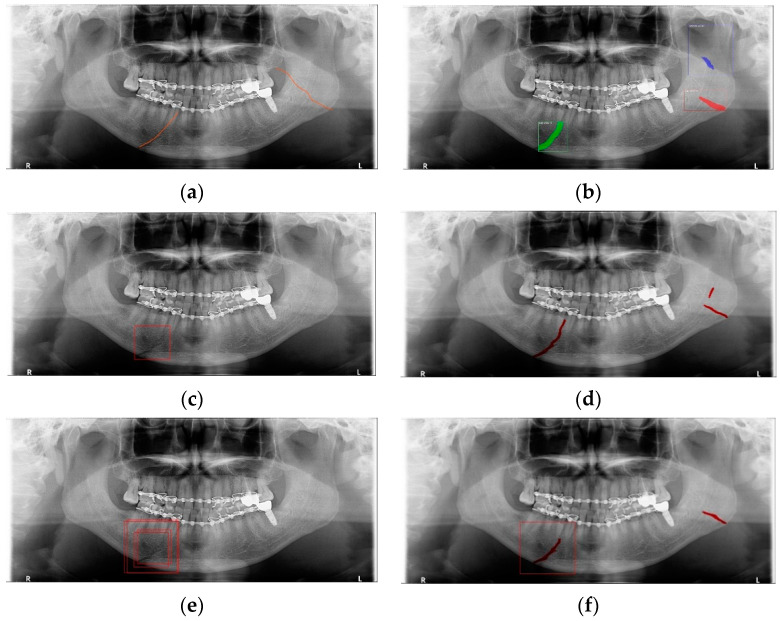
The comparison of mandibular fracture detection: (**a**) oral and maxillofacial radiologist, (**b**) Mask R-CNN (with colored polygons and boxes), (**c**) YOLOv4 (with a red box), (**d**) U-Net (with red lines), (**e**) LAT YOLOv4 (with red boxes), and (**f**) U-Net with LAT YOLOv4 (with a red box and lines).

**Figure 14 life-12-01711-f014:**
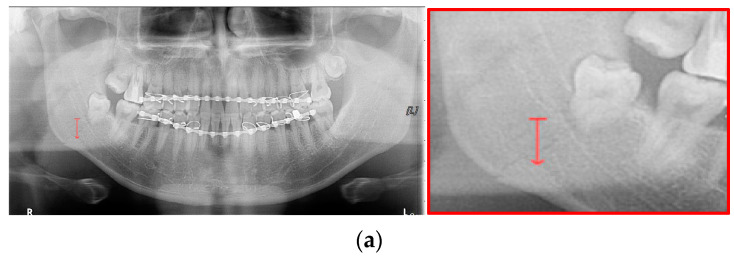
The comparison of SLAT and MLAT processing with line profiler: (**a**) normal radiograph, (**b**) SLAT-processed, and (**c**) MLAT-processed.

**Figure 15 life-12-01711-f015:**
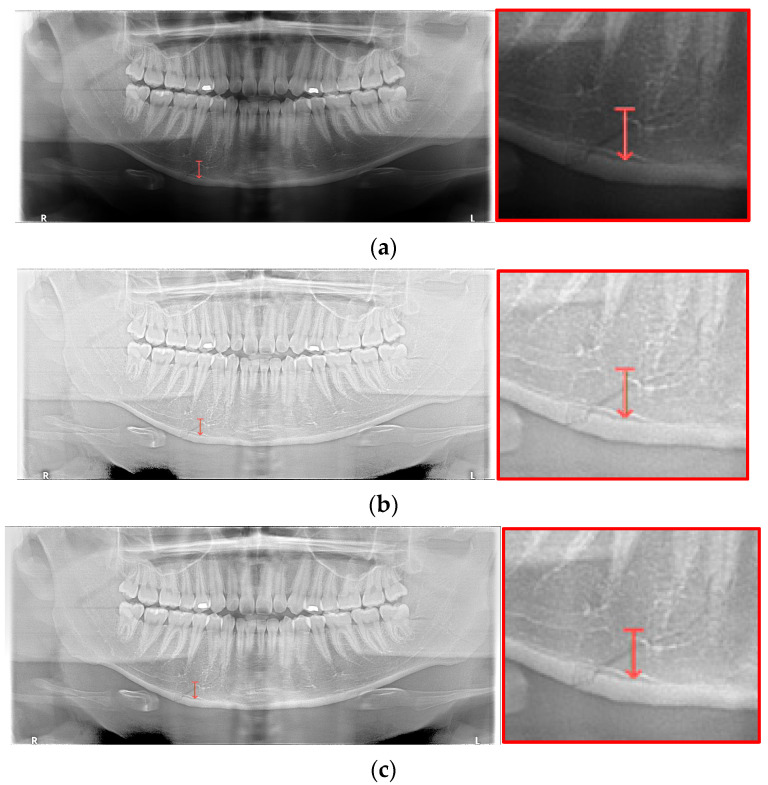
The comparison of SLAT and MLAT processing with line profiler: (**a**) normal radiograph, (**b**) SLAT processed, and (**c**) MLAT processed.

**Figure 16 life-12-01711-f016:**
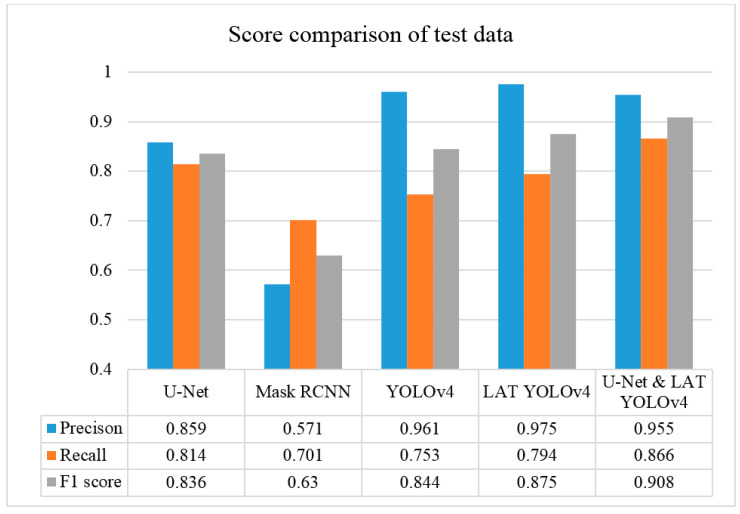
The score comparison of U-Net, Mask R-CNN, YOLOv4, LAT YOLOv4, and U-Net with LAT YOLOv4 modules.

**Figure 17 life-12-01711-f017:**
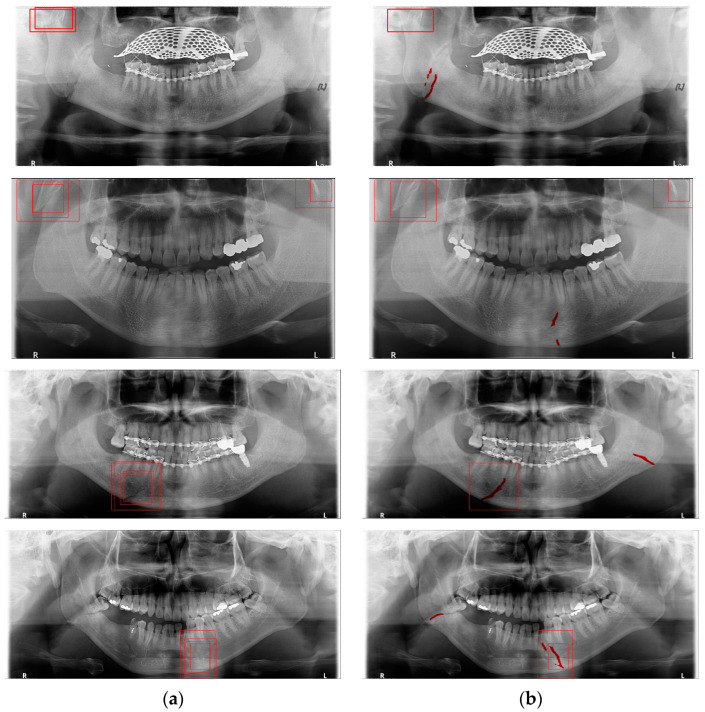
The comparison of LAT YOLOv4 and U-Net with LAT YOLOv4 detection results. (**a**) LAT YOLOv4 (with red boxes), and (**b**) U-Net & LAT YOLOv4 (with red boxes and lines).

**Table 1 life-12-01711-t001:** Comparison of three activation functions in U-Net.

	U-Net (ReLU)	U-Net (Leaky ReLU)	U-Net (ELU)
Precision	0.859	0.828	0.916
Recall	0.814	0.794	0.784
F1 Score	0.836	0.811	0.844

**Table 2 life-12-01711-t002:** Comparison with YOLOv4 modules.

	YOLOv4	LAT YOLOv4
Precision	0.961	0.975
Recall	0.753	0.794
F1 score	0.844	0.875

**Table 3 life-12-01711-t003:** Comparison with U-Net modules.

	U-Net (ELU)	U-Net (ELU) with Random Crop
Precision	0.916	0.917
Recall	0.784	0.794
F1 score	0.844	0.851

**Table 4 life-12-01711-t004:** The fracture distribution of training data.

	Symphysis	Body	Angle	Ramus	Condyle	Coronoid	Total
Number of fractures	153	145	183	91	39	13	624
Percentage	24.519%	23.237%	29.327%	14.583%	6.25%	2.084%	100%

**Table 5 life-12-01711-t005:** The fracture distribution of test data.

	Symphysis	Body	Angle	Ramus	Condyle	Coronoid	Total
Number of fractures	25	20	29	8	13	2	97
Percentage	25.773%	20.619%	29.897%	8.247%	13.021%	2.063%	100%

**Table 6 life-12-01711-t006:** The parameters of the LAT YOLOv4 and U-Net modules.

Option	LAT YOLOv4	U-Net
Batch size	64	6
Subdivision	16	-
Resolution	608×608	512×512
Learning rate	0.0001	0.0001
Epoch	711 (12,000 max batches)	300

**Table 7 life-12-01711-t007:** The line profile data of Figure 14 and Figure 15.

	Maximum Pixel Value	Minimum Pixel Value	Average Pixel Value	Standard Deviation
Figure 14a	180	156	169.8	5.6
Figure 14b	230	206	217.1	5.4
Figure 14c	217	196	208.4	5.0
Figure 15a	98	53	68	9.3
Figure 15b	222	145	171.2	14.6
Figure 15c	242	166	194.4	15.2

**Table 8 life-12-01711-t008:** The comparison of processing time between LAT YOLOv4 and the proposed module.

Processing Section	U-Net	U-Net Tooth Area	U-Net False-Detection	U-Net Total Result	YOLO MLAT	YOLO SLAT	YOLO Total Result	TOTAL TIME	Total Time/One Image
LAT YOLOv4	**-**	**-**	**-**	**-**	1.439	1.438	4.956	7.833	0.131
Proposed model	8.258	9.526	0.46	5.231	1.439	1.438	4.956	31.308	0.522

## Data Availability

Not Applicable.

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
