# Peer review of "Combined Deep Learning Techniques for Mandibular Fracture Diagnosis Assistance"

_life, 2022, doi:10.3390/life12111711_

Round 1

Reviewer 1 Report

The article is very well written. It is an advance of previous work by the same authors.

In the conclusions, it is stated that the proposed method diagnoses mandibular fractures in panoramic images through deep learning, reducing the cost and time. How much cost and time are you talking about?

It is requested to make available the source code developed and used in the tests. 

It is recommended that this work, published this year, be referenced:

Wang, X., Xu, Z., Tong, Y., Xia, L., Jie, B., Ding, P., ... & He, Y. (2022). Detection and classification of mandibular fracture on CT scan using deep convolutional neural network. Clinical Oral Investigations, 1-9.

Author Response

We attached the reply letter. Thank you so much for kind review.

Reviewer 2 Report

In this manuscript the authors demonstrate improved accuracy in mandibular fracture detection over their previous efforts in this area (Ref. 7), by combining the predictions of two different networks. They train a U-net based line segmentation model that shows better performance in segmenting irregularly shaped features, over their previously reported LAT YOLOv4 network (from Ref. 7). On the other hand, this LAT YOLOv4 model had better detection performance on shear or displaced fractures. By summing these two network outputs, the authors observe increased detection accuracy compared to the individual network outputs.

Major comments

  1. Overall, I am quite concerned about the LAT YOLOv4 model performance shown in this work. First, the LAT YOLOv4 model is able to capture the angle fracture in Fig. 11e while missing it completely in Fig. 13e. Second, in Fig. 12c, YOLOv4 is able to capture the angle fracture whereas the supposedly improved model LAT YOLOv4 misses it completely in Fig. 12e. Thus, it appears that the LAT YOLOv4 model may not have been trained to optimality. Consequently, I’m skeptical if proper training of this model may achieve similar performance even without adding the U-Net, which by the way, is the only novelty in this work (compared to Ref. 7) 

  1. Can the authors comment on how similar the test and training datasets are? In a real world scenario, one cannot assume that the test data is from the same distribution as the training data which is why a well-generalizable model is key to real world deployment. The authors should rigorously demonstrate model generalizability by running inference on edge cases or out-of-distribution data points.

  1. The language in the paragraph from lines 159-167 is extremely confusing and potentially inconsistent with the results of Fig 3(d). It is unclear whether the model used for generating Fig 3(d) was trained with shear and dislocated fractures. If the U-net was trained without shear fractures, then how can we expect it to catch shears anyway? If the authors indeed included shear structures in the training set, they must clarify the annotation strategy (bounding box or polygons) used, since they claim that “shear fractures or displaced fractures are difficult to label as U-Net training data” in line 161.

  1. The authors only focus on accuracy throughout this manuscript. To justify the word  “diagnosis assistance” in the title, they should also clarify model inference time and/or throughput requirements in a hospital setting, especially since their final model (two U-Nets + one LAT YOLOv4) is significantly more complicated than a single network like the Ref. 7 version of LAT YOLOv4.

Minor comments

1. It looks like the tooth-line-exclusion model described schematically in Fig. 7 can be integrated into the line-segmenter U-Net, by possibly adding the tooth exclusion label as an ignore mask in the U-Net loss function. Can the authors comment on why they introduce this additional extraction module which can potentially sacrifice training and inference speed?

2. For LAT YOLOv4, please specify the class distribution in the training images, ie. how many images per class and discuss any potential class imbalance.

3. Please clarify whether the results in Fig. 14 are demonstrated for train data or test data.

4. In lines 259-260, please clarify if rotation augmentation was used.

5. Language that significantly hurts readability (and must be improved)

  • Lines 159-163

  • Line 252, 253

  • Lines 365

Author Response

(The authors gave the same response as above.)

Round 2

Reviewer 2 Report

Reviewer comments are directly written into the (attached) author response. These comments are marked in cyan and italicized. 

Author Response

We attached the reply report. Thank you so much.
